# Dynamic Performance of Laminated High-Damping and High-Stiffness Composite Structure Composed of Metal Rubber and Silicone Rubber

**DOI:** 10.3390/ma14010187

**Published:** 2021-01-02

**Authors:** Xiaoyuan Zheng, Zhiying Ren, Liangliang Shen, Bin Zhang, Hongbai Bai

**Affiliations:** 1School of Mechanical Engineering and Automation, Fuzhou University, Fuzhou 350116, China; zhengxycxf@163.com (X.Z.); llshenlyndon@126.com (L.S.); omaxuzb@163.com (B.Z.); bhb11.fzu@gmail.com (H.B.); 2Engineering Research Center for Metal Rubber, Fuzhou University, Fuzhou 350116, China

**Keywords:** metal rubber, silicone rubber, laminated composite damping structure, embedded interlocking structure, periodic cyclic excitation, damping energy dissipation

## Abstract

In this study, a laminated composite damping structure (LCDS) with metal rubber (MR) as matrix and silicone rubber (SR) as reinforcement has been designed. The embedded interlocking structure formed by the multi-material interface of the LCDS can effectively incorporate the high damping characteristics of traditional polymer damping materials and significantly enhance the adjustable stiffness of the damping structure. Based on the periodic cyclic vibration excitation, dynamic tests on different laminated structures were designed, and the damping performance and fatigue characteristics under periodic vibration excitation of the LCDS, based on MR and SR, were explored in depth. The experimental results exhibited that, compared to single-compound damping structures, the LCDS with SR as reinforcement and MR as matrix has excellent stiffness and damping characteristics. The incorporation of the silicon-based reinforcement can significantly improve the performance of the entire structure under cyclic fatigue vibration. In particular, the effects of material preparation and operating parameters on the composite structure are discussed. The effects of MR matrix density, operating frequency, amplitude, and preload on the stiffness and damping properties of the MR- and SR-based LCDS were investigated by the single factor controlled variable method. The results demonstrated that the vibration frequency has little effect on the LCDS damping performance. By increasing the density of the MR matrix or increasing the structural preload, the energy dissipation characteristics and damping properties of the LCDS can be effectively improved. With the increase in vibration excitation amplitude, the energy consumption of the LCDS increases, and the average dynamic stiffness changes at different rates, resulting in the loss factor decreasing first and then increasing. In this study, a damping structure suitable for narrow areas has been designed, which overcomes the temperature intolerance and low stiffness phenomena of traditional polymer rubber materials, and provides effective guidance for the design of damping materials with controllable high damping and stiffness.

## 1. Introduction

Vibration and noise reduction through damping materials has always been an important research direction of damping technology [1]. In recent years, with the deepening of research on vibration reduction equipment, damping materials have become diversified to meet various complex environmental and vibration reduction requirements under different conditions. In general, existing damping materials can be approximately classified into four categories: viscoelastic, composite, intelligent, and metal damping materials. Among these, viscoelastic damping materials are currently the most commonly used [2]. In order to improve the damping properties of these materials and broaden their applicable temperature range [3], a great amount of research has been conducted concerning rubber blending, copolymerization, organic small molecule hybridization damping, filler modification, solution coprecipitation, and interpenetrating polymer networks (IPN) technology [4]. Silicone rubber (SR) is favored in engineering applications due to its high-temperature resistance. Ren [5] used SR as the base material, and boron nitride and short carbon fibers as the filler material to develop a high-temperature and high-damping silicone rubber composite; however, its load-carrying capacity still needs to be improved.

In addition to the abovementioned methods, some scholars have designed multi-layer [6] and restraint damping structures [7] from the structural design point of view, achieving certain results. Xiao [8] prepared a super elastic NiTi alloy/polyurethane constrained structure to be used as a damping composite material using a hyper-elastic NiTi alloy as the constraint layer. Through comparative verification of the single layer/multi-layer damping layer structures, the damping performance, interlayer bonding strength, and bending properties of the material were investigated. Taking the subway test section as the engineering background, He [9] analyzed the density, drying time, static and dynamic mechanical properties, and solid content of materials, and discussed the constrained damping structural performance of different material layers based on a multi-layer constrained damping ceramic tile plate. Liang [10] designed a new five-layer sandwich composite instrument panel with advanced damping characteristics and specific stiffness characteristics. Based on the theory of multi-layer damping composite design, Yang [11] prepared a multi-layer composite damping structure using traditional polymer rubber as the matrix, and investigated the effect of damping layer thickness, material loss factor, and elastic modulus on structural performance. Li [7] theoretically analyzed the effect of material anisotropy and structural components on the characteristics and control mechanism of layered damping structures. The results exhibited that the effect of the flexible layer on damping is greater than that of the stress coupling layer. Compared to changing the material components, the damping performance of the materials optimized by structural design based on the existing damping materials exhibited simple and efficient characteristics and was found to be more practical. However, compared to traditional high damping materials, the demand for large stiffness support of damping structures in industrial production is increasing continuously [12]. Therefore, many scholars have turned their attention to the design of multi-layer composite damping structures based on metal rubbers (MRs), which are the most typical metal damping materials.

MR is a new elastic porous material which is named for its characteristics of high elasticity and large damping. It is manufactured, as shown in Figure 1a, by the cold stamping of metal wires [13]. Based on the pressing process, various structures and shapes can be prepared. Due to the raw materials of the wires, MR has the ability to adapt to various complex environments. The metal wires inside the MR are interlocked, forming a spatial network structure (Figure 1b). Under the action of external alternating load, the extrusion, sliding, and friction between the metal wires are the main ways to achieve damping and vibration reduction. Some studies have shown that the air damping effect of MR may consume a lot of energy under high-speed deformation, and it has a good development prospect in the field of impact resistance [14]. In addition to damping performance, the special molding process provides MRs with excellent stiffness performance. Figure 1c illustrates the application of laminated MR in pipeline vibration reduction. Xiao [15] designed a laminated MR coating vibration reduction structure for high-temperature pipeline vibration reduction and introduced the variation law of its energy consumption characteristics based on parameters such as amplitude, frequency, ambient temperature, and density experimentally. Ao [16] designed an MR damping pad to solve the vibration reduction problem of an engine pipeline. They conducted experiments to discuss the effect of different molding densities, component molding technologies, installed pre-compression shrinkage, and excitation force level on the damper. All metal damping materials are made of single materials. Despite the design of the layered structure, the damping performance is relatively low and cannot further meet the complex working conditions of periodic fatigue vibration. To improve the damping performance of MRs, Lu [17] designed a new damper, based on MR and SR. The dynamic characteristics were experimentally investigated, and a dynamic model was established based on the experimental results. However, in the above study, only the tangential energy consumption of the laminated damping structure was discussed, while the stiffness performance and load-bearing damping capacity of the laminated multi-base damping structure were not analyzed in depth.

In this study, a laminated composite damping structure (LCDS) of MR and SR is designed based on anisotropic physical interface characteristics, which not only meets the high damping requirements, but also ensures that the structure has high stiffness and performance stability characteristics. This kind of structure just stacks different materials together, instead of bonding different materials by chemical or physical methods. Based on the dynamic response of the LCDS under interactive load, the stiffness, damping energy dissipation, and fatigue stability characteristics of the LCDS are explored in depth. The effects of frequency, amplitude, MR matrix density, and structural preload on LCDS performance are investigated by single factor control tests. The design of the MR- and SR-based LCDS can provide an effective reference for the design of damping materials with high stiffness and high damping characteristics in the narrow-spaced structure system under complex conditions.

## 2. LCDS Preparation Based on MR and SR

In this work, LCDS is a composite damping structure formed by combining long strips of MR and SR through superposition. As can be seen in Figure 2, the LCDS is formed by (a) selecting proper metal wires and preparing a spiral coil, (b) constant pitch stretching of the spiral coil, (c) winding of the blank to form the MR blank, (d) flat MR preparation by cold stamping technology in a special mold, (e) post-treatment, and (f) lamination with SR. The synergistic effect of the different laminated structures and the extrusion-based physical contact interface between the components provides the structure with excellent damping stiffness characteristics that differ significantly from those of the single materials.

### 2.1. Material Selection and Spiral Winding

The selection of the wire material plays an important role in the overall stiffness, elastic recovery force, and damping loss characteristics of the structure, since it is the basic winding unit of MR [18]. In this study, 304 austenitic stainless-steel wire (1Cr18Ni9Ti) (Tiancheng Stainless Steel Products Co., Ltd., Xinghua, China) was used as the MR matrix material (wire diameter: 0.1–0.3 mm) in the LCDS due to its excellent environmental adaptability and life stability. The chosen straight metal wire was transported into a special equipment, and a spiral coil with a certain diameter (2–10 mm) was wound with space spiral orbit [19].

### 2.2. Constant Pitch Stretching and Winding of the Blank

The spiral coil was stretched at a fixed pitch (2–10 mm) and wound on the blank mandrel at a certain angle (30–60°) using numerical winding equipment. This is a key step in the preparation of the MR, which produces a significantly different wire turn, interpenetrating the network structure through different pitch or winding angles [20].

### 2.3. Stamping and Heat Treatment

As shown in Figure 3, the blank was placed in the stamping die in the THD 32–100 four-column hydraulic press (Huidian Electromechanical Equipment Development Co., Ltd., Tianjing, China), and was cold formed according to the limited tonnage. The plastic deformation stability of the material was ensured under the maximum stamping force for 30 s. In general, after the mold was removed, to eliminate the effect of internal residual stress developed after molding, stress annealing and other heat treatment processes were used to ensure the performance stability of the MR [21]. The forming parameters of the MR materials in this study are shown in Table 1.

### 2.4. Silicone Rubber Selection and Laminated Composite Structure

Among traditional polymer rubber materials, SR has been widely used in various engineering applications, especially in high-temperature environments, due to its excellent high-temperature resistance and stable mechanical properties in a wide temperature range [22]. Therefore, in this work, SR and MR were used as reinforcements/matrix to prepare the LCDS (Figure 4). Considering the contact layer interface and the industrial demand for vibration damping structures in narrow idle space, the specimens were shaped as thin plates. The selected parameters of SR and metal wires are shown in Table 2.

## 3. Experimental Procedure

### 3.1. Specimen Preparation

The preparation parameters of the MR are listed in Table 3. Based on the preparation of the thin MR (M) plates and SR (S) (Figure 5), different composite damping structures were formed by combining the laminated specimens (Figure 6). The structure was composed of three layers. Each layer was 4 mm thick, and the total thickness of the LCDS was 12 mm.

The selection of different components to serve as matrix and reinforcement in the composite structure is the key factor affecting LCDS properties, which, at the same time, introduces different physical contact interfaces in the LCDS. As shown in Figure 6, MR and SR formed four different laminated structures: (a) M-M-M, (b) S-S-S, (c) S-M-S, and (d) M-S-M. In particular, the density has a unique effect on MR stiffness and damping characteristics [23]. Therefore, MR specimens with different densities were prepared to investigate the LCDS damping energy dissipation characteristics. The specimen parameters are given in Table 1.

### 3.2. Dynamic Test Based on Sinusoidal Excitation

Dynamic tests on the LCDS based on MR and SR were carried out on an SDS-200 (Sinotest Equipment Co., Ltd., Changchun, China) testing machine under sinusoidal excitation to investigate the dynamic mechanical properties and damping characteristics of the prepared composite materials. The main parameters of the testing system are listed in Table 4. Figure 7 illustrates a schematic diagram of the testing system. It can be seen that the upper and lower connecting rods were, respectively, connected with the upper and lower pressing plates through threaded bolts. Before the test, the upper and lower connecting rods were connected with the upper and lower clamps of the SDS-200 dynamic and static universal testing machine, maintaining the upper and lower platens parallel. During the entire experiment, the lower platen was fixed, while the upper chuck was controlled to move up and down at a certain frequency and amplitude by the controller to produce compression deformation on the specimen. During the experiment, the sample is placed on the lower platen and moves down through the beam of the equipment, so that the upper platen preloads the sample with a certain displacement. The position at this time is the zero point. When moving downward from this zero point, the pressure is negative, and when moving upward from the equilibrium point, the pressure is positive. The sample only compressed during the whole experiment. The test element was rectangular, while the upper and lower platens were square. In the test, the centerline of the rectangular element is placed along the diagonal line of the positive pressing plate, while the four sides of the upper and lower square pressing plates were parallel, ensuring that the force on the component was radial and the effect of the axial force was eliminated.

In the dynamic test with sinusoidal displacement as the excitation load, the hysteresis loop directly reflects the load deformation correlation, and the energy dissipation capacity of the structure during loading and unloading can be obtained. In addition, the hysteresis loop reflects the damping characteristics of the material, and the area surrounded by the hysteresis loop represents the energy ΔW absorbed by the structure in one period. In this study, the restoring force signal of the LCDS under sinusoidal excitation was recorded, and the energy dissipation and maximum elastic potential energy for a stable period were obtained. Finally, the equivalent energy dissipation factor of the LCDS was calculated to effectively characterize the damping characteristics of the structure. Figure 8 demonstrates the hysteresis loop, the energy dissipation in a stable period, and the maximum elastic potential energy of the system.

The displacement excitation applied by the testing machine can be calculated as follows:(1)$$X=X_{0}\cos(wt+\alpha )$$
where α is the initial phase, $X_{0}$ is the displacement amplitude, and W is the displacement value loaded in Equation (1), which can be discretely expressed as:(2)$$X_{i}=X_{0}\cos(\frac{2\pi i}{N}+\alpha ),i=1,2,\cdots ,N$$
where N is the number of sampling points in one vibration period, and $N=f_{0}/f$, where $f_{0}=2500\;\mathrm{Hz}$ represents the sampling frequency and f is the loading frequency.

Accordingly, the energy ΔW dissipated in one period can be expressed as:(3)$$\begin{array}{ll} \Delta W & =\oint FdX\\ & =\oint Fd\left[X_{0}\cos(\omega t+\alpha )\right]\\ & =-\frac{2\pi X_{0}}{N}\sum_{i=1}^{N}F_{i}\sin(\frac{2\pi i}{N}+\alpha ) \end{array}$$
and the maximum elastic potential energy stored in the material is:(4)$$\begin{array}{ll} W & =\frac{1}{2}\times \frac{F_{\max}-F_{\min}}{2X_{0}}\times X_{0}^{2}\\ & =\frac{1}{2}\times \bar{K}\times X_{0}^{2} \end{array}$$
where $F_{\max},F_{\min}$ are, respectively, the maximum and minimum values of the collected restoring force data, and $\bar{K}$ represents the average dynamic stiffness, which can be calculated by $\bar{K}=\frac{F_{\max}-F_{\min}}{2X_{0}}$.

The equivalent loss factor of the material can be obtained as follows:(5)$$\begin{array}{ll} \eta & =\frac{\Delta W}{2\pi W}\\ & =-\frac{4f}{f_{0}\left(F_{\max}-F_{\min}\right)}\sum_{i=1}^{N}F_{i}\sin(\frac{2\pi i}{N}+\alpha ) \end{array}$$

The equivalent loss factor η characterizes the dynamic damping effect of a structure under periodic excitation. The smaller its value, the better the damping characteristics of the structure. However, under the continuous action of periodic dynamic load, the damping structure will undergo fatigue damage, which will alter its stiffness and damping performance. Therefore, the long-term dynamic performance of the damping structure can be accurately expressed by the stiffness damage factor:(6)$$D(n)=\frac{\bar{K}(0)-\bar{K}(n)}{\bar{K}(0)}=1-\frac{\bar{K}(n)}{\bar{K}(0)}$$
where D(n) represents the stiffness damage factor, $\bar{K}$ represents the average dynamic stiffness, $\bar{K}(0)$ represents the initial stiffness of the damping structure, and $\bar{K}(n)$ represents the structural stiffness of the structure subjected to n sinusoidal vibrations.

The boundary condition for the failure factor is as follows:(7)D(n=0)=0
while the damage criterion is:(8)$$D\geq D_{r}$$
where $D_{r}$ is the critical damage factor. When $D=D_{r}$, the total number $N_{f}$ of vibrations the material is subjected to is the fatigue life.

## 4. Results and Discussion

### 4.1. Effect of Laminated Structure on Damping Characteristics

#### 4.1.1. Dynamic Damping Characteristics of Different Laminated Structures

Periodic loading tests on different LCDSs under positive excitation were carried out. In each group of tests, the displacement was controlled during loading and unloading cycles with equal amplitude to obtain the curves representing the relationship between structural force and excitation load. In the experiments, the loading amplitude was 0.5 mm, and the loading frequency was 1 Hz. In all groups, the preload remained unchanged. The hysteretic curves of the different laminated structures were drawn based on the acquired experimental data, as shown in Figure 9. The damping characteristics of the different laminated structures were obtained quantitatively, based on Equations (3)–(5) (Table 5).

As can be observed in Figure 9, the dynamic restoring force (stiffness) of the M-M-M laminated structure under different densities was low. The dynamic stiffness and energy dissipation capacity of the LCDS increased gradually with increasing MR density. This was attributed to the reduction in the micro motion space of the wire turns in the material and the increased contact friction between wire turns, which finally exceeded the corresponding values of the S-S-S damping structure. At the same time, as can be quantitatively seen in Table 5, the loss factor presented a downward trend with increasing MR density, and was finally lower than the corresponding value of the SR. This indicated that its energy consumption characteristics were gradually improved. Compared to this, the effect of density on MR average dynamic stiffness was more significant.

It can clearly be observed in Figure 9 that the energy dissipation capacity of the multi-material composite damping structures (M(80)-S-M(80) and S-M(80)-S) was better than that of the single-compound damping structures (S-S-S and M(80)-M(80)-M(80)). Compared to the single-compound damping structures, the contact interface between the different laminated components in the multi-component composite damping structure made the entire structure generate a synergistic effect. The metal turns on the MR surface and the SR formed an embedded interlocking structure (EIS). As can be seen in Figure 10b, due to the synergistic effect of the MR wire turns “locking” on the outer surface of the SR, the angle between the deformed wire turns at the depression of the SR surface and the surrounding wire turns increased under dynamic load excitation. Therefore, the friction force required for relative sliding was increased, thus improving the dynamic stiffness and energy consumption of the LCDS overall structure. Consequently, the MR damping performance can be significantly improved by the MR- and SR-based laminated composite structure.

Furthermore, it can be noticed in Figure 9 that, compared to M(80)-S-M(80), S-M(80)-S exhibited better damping energy dissipation characteristics. According to Figure 11, the matching of MR and SR was equivalent to the embedded contact between MR and flexible body. Under the action of load, the high-hardness wire turns are embedded in the low-hardness SR. The embedded interlocking structure formed between the contact interfaces can effectively improve the state of MR under forced deformation. At the same time, compared to the M-S-M structure, the LCDS composed of S-M-S had a larger effective contact area when excited under load F. The increased effective bearing area makes the composite damping structure, with the SR surface layer, transfer load more evenly, while having more advanced vibration and energy dissipation characteristics under excitation.

#### 4.1.2. LCDS Fatigue Characteristics

In engineering applications of damping materials, the damping structure is often required to maintain stable damping performance under periodic cyclic excitation; thus, studying the structural fatigue characteristics is of high importance. Currently, there is little research on the fatigue characteristics of LCDS composed of MR and SR. In the fatigue testing experiment of this study, the excitation amplitude was 0.5 mm and the loading frequency was 5 Hz. The type of metal rubber used in the fatigue test is M (80). The experimental results are shown in Figure 12.

It can be seen in Figure 12a,b that the energy consumption curves of the single-material composite damping structures (S-S-S, M-M-M) exhibited a relatively stable trend with the increase in loading cycles. The energy consumption and energy dissipation factors of the S-M-S were generally better than those of the single-compound damping structures. The energy consumption and energy dissipation factors of the LCDS based on S-M-S tended to be stable with increasing number of cycles, which reflected that the structure is able to maintain a relatively stable damping performance under periodic load.

The damping effect of MR is attributed to the squeezing friction motion between the inner MR turns under interactive load (Figure 13b). Moreover, the instability and fatigue damage of the wire turns initial running in stage can often lead to performance instability characteristics under periodic cyclic load, which was specifically shown in the evolution process of the energy dissipation factor of the M-M-M structure under periodic load (Figure 12b). The damping characteristics of the SR are mainly due to the friction between molecular chains. More specifically, due to the physical entanglement between the molecular chains of the SR, the thermal motion between molecular segments is enhanced with increasing number of load cycles, while the entanglement effect is weakened. This makes the loss factor decrease first and then increase, and reach a relatively stable state after a certain number of cycles (S-S-S curve; Figure 12b). In the S-M-S composite damping structure, the performance instability caused by the MR friction damage instability effect can be significantly improved on the contact interface of the physical layer of the structure. Due to that, the stress distribution mode of the entire structure is effectively achieved through the mutual locking effect of the EIS generated between MR and SR (Figure 13a). The loss factor of the S-M-S structure tended to be stable under periodic load, without sudden changes in the damping characteristics. As shown in Figure 13c, a certain amount of debris and metal broken wire are produced under periodic load. Due to the friction of the metal wire, the surface wear is formed on the surface of SR, as shown in Figure 13d. At the same time, it can be seen that the depth of wear is not very deep.

Figure 12c displays the damage factor curves of the different laminated structures under different numbers of load cycles. The smaller the damage factor D, the longer the life of the structure. As can be seen in Figure 12c, under cyclic loading, the damage factor of the S-S-S structure fluctuated stably, while those of the M-M-M and S-M-S structures increased continuously. Among them, the damage factor of the M-M-M structure exhibited a rapid growth trend, due to the running in process of internal wire turns in the initial stage, which indicates that the SR material has a longer service life than the MR material. Therefore, the SR laminated structure can be introduced into the MR material to effectively improve its fatigue life, while ensuring the high-stiffness performance of the damping structure.

### 4.2. Single-Factor Control Experiments and Result Analysis

In the work described above, the dynamic damping performance and fatigue characteristics of different laminated structures were investigated, and it was concluded that the S-M-S LCDS can effectively improve the damping performance and fatigue characteristics of materials, in order to meet the industrial requirements for large stiffness vibration reduction structures. However, MR, when used as the matrix of composite structures, comes with complex preparation parameters and application conditions that play an important role in the overall performance of the structure. Therefore, the effect of matrix density, load amplitude, frequency, and preload on the mechanical properties of the S-M-S composite damping structure was investigated in detail. Table 6 provides the dynamic experimental parameters based on single-factor variable control. To eliminate the error caused by the surface roughness of the material, at least 1.5 mm preload should be applied before testing.

#### 4.2.1. Effect of MR Matrix Density on Damping Performance

To study the effect of matrix density on the S-M-S composite damping structure, five MRs with different densities were selected as the matrix for periodic dynamic contrast experiments. The test parameters are listed in Table 6, and the experimental results are given in Figure 14 and Table 7.

It can be seen in Figure 14 that, due to the sharp increase in the number of contact points and contact friction pairs in the MR matrix with increasing density, the overall dynamic stiffness of the LCDS increased, and the area of the hysteresis curve increased significantly, showing that the damping and energy dissipation performance was enhanced. According to Table 7, with the increase in MR matrix density, the average dynamic stiffness $\bar{K}$ and energy consumption ΔW of the S-M-S laminated composite structure exhibited a significant upward trend. However, due to that the increase rate of the average dynamic stiffness was greater than that of the energy consumption ΔW, the energy dissipation factor of the S-M-S damping structure increased first and then decreased. Therefore, in laminated composite materials, the loss factor of the structure can be effectively controlled by adjusting the density of the MR matrix, ensuring that the vibration reduction characteristics of a structure in engineering applications will be preserved.

#### 4.2.2. Effect of Amplitude on Damping Performance

In order to investigate the effect of periodic excitation amplitude on the performance of the S-M(80)-S composite damping structure, periodic dynamic experiments with an amplitude ranging between 0.2–1 mm were performed. The preload is 2.0 mm and test parameters are given in Table 6, and the experimental results are shown in Figure 15.

As can be seen in Figure 15a and Table 8, the area of the hysteretic curve of the damping structure increased, the corresponding energy dissipation capacity increased, and its dynamic recovery capacity decreased with increasing amplitude. As can be observed in Figure 15b,c, when the periodic vibration amplitude was increasing, the dynamic restoring force and energy dissipation capacity of MR exhibited an increasing trend. This is due to the fact that the increase in amplitude makes the spiral coil beams that are in contact with each other slide sufficiently. In addition, the large amplitude of vibration deformation leads the number of contact points and the amount of friction slip of inter turn to increase significantly, which leads eventually to an increase in the energy consumption of the M-M-M damping structure. At the same time, the fretting space of the wire turns in the MR is continuously compressed with increasing periodic amplitude, and several viscous extrusion contact phenomena appear when the spiral coil turns are further compressed, leading to an increase in the dynamic stiffness.

It can be seen in Figure 15b that the dynamic stiffness of S-S-S and S-M-S decreased with increasing amplitude. However, due to the embedded interlocking structure formed between the S-M-S heterogenous components, there is a physical layer of embedded wire turns and hooked structures between MR and SR, which increases the reaction force of the S-M-S under load, and its dynamic stiffness is also significantly greater than that of traditional MR materials. The damping properties of SR, being a polymer material, originate from the friction between molecules. The increase in amplitude increases the friction between molecules, and the introduction of SR significantly improves the energy dissipation capacity of the S-M-S composite damping structure (Figure 15c).

According to Table 8, the loss coefficient of the damping structure decreased first and then increased with increasing amplitude. This is attributed to the different relative change rate of the energy consumption capacity and the dynamic recovery capacity. Moreover, it can be seen in Figure 15b,c that the S-M-S composite damping structure with small amplitude exhibited higher average dynamic stiffness and energy dissipation performance than the single-compound damping structure, better meeting the load bearing requirements.

#### 4.2.3. Effect of Frequency on Damping Performance

In order to investigate the effect of periodic excitation frequency on the S-M(80)-S composite damping structure, periodic dynamic experiments at frequencies ranging between 1–9 Hz were performed. The test parameters are given in Table 6, and the test results are presented in Figure 16.

Figure 16a shows that the dynamic restoring force and energy dissipation capacity of the LCDS were less affected by frequency. According to Table 9, the loss factor η, average dynamic stiffness, and energy dissipation of the S-M-S laminated composite damping structure first decreased and then increased with increasing vibration frequency. This is attributed to the physical entanglement between the molecular chains in the SR, and to that the entanglement force in the movement of the molecular chains under periodic cyclic loading must be overcome [24]. With the increase in loading frequency, the number of movements between molecular chains in unit time increased, and the increase in thermal motion weakened the entanglement effect. The activities of segments, links, and side bases were easier than the initial ones; thus, the average dynamic stiffness of the silicone rubber increased first and then decreased. In addition, the entanglement effect of the molecular chains in the SR tended to increase to decrease, which also makes the motion state of molecules inside the material gradually weaken, with the increase in frequency, causing the energy consumption of intermolecular friction to increase after a small decrease in unit time. On the other hand, the increase in the low frequency vibration will lead to loss of synchronization between the internal wire slipping frequency and the loading frequency, and the insufficient wire sliding will weaken the energy consumption effect. In contrast, the MR itself is a porous structure. With the increase in loading frequency, the frequency of internal air compression, the friction between air and metal wire, and the energy consumption increase. Consequently, the S-M-S damping performance decreases first and then increases with increasing vibration frequency (Figure 16b,c).

#### 4.2.4. Effect of Preloading on Damping Performance

Amplitude and preload are the main factors affecting porous materials, and there is a certain correlation. However, in the actual application process, it is often given a certain preload to ensure the stability of the system, and then the actual applied amplitude is considered [25]. Therefore, we discuss the influence of amplitude and preload on the LCDS separately. In order to study the effect of preload displacement on the S-M(50)-S composite damping structure, periodic dynamic experiments of different preload magnitudes, ranging between 1.5–3.5 mm, were performed. The test parameters are listed in Table 6, and the test results are presented in Figure 17.

It can be seen in Figure 17a that, with the increase in preload, both the wire turn micro-element contact pairs in the MR and the molecular chain movements in the SR exhibited an increasing trend, which led to a gradual increase in the dynamic recovery capacity and energy dissipation capacity of the damping structure.

According to Table 10, the loss factor of the S-M-S structure increased first and then decreased with increasing preload. In addition, the results demonstrated that, with the increase in preload, the total energy consumption increased, and the MR damping component exhibited nonlinear stiffness characteristics, which can be divided into three types: linear elastic, soft, and hard [26]. This made the stiffness of the S-M-S structure present different growth rates with the increase in preload. Combined with Equations (3)–(5), it can be found that different relative growth rates of the average dynamic stiffness and energy consumption make the loss factor increase first and then decrease. Furthermore, it can be seen in Figure 17b,c that the stiffness and damping performance of the S-M-S damping structure were better than those of the single damping component structures (S-S-S, M-M-M) only when the density of the MR reached a certain value. Based on the S-M-S average dynamic stiffness and energy consumption change rate, it can be deduced that, when the core density is low, the stiffness range of the S-M-S structure can be wide and can be altered by simply changing the preload.

## 5. Conclusions

LCDSs, based on MR and SR, were designed, and the dynamic stiffness and damping energy dissipation characteristics of the composite damping structure under periodic sinusoidal excitation were investigated. The designed structure meets the requirements of excellent damping performance, while the high stiffness and fatigue stability of the material are ensured. The details are as follows:

(1) Composite damping structures with different laminated compounds were designed, and their dynamic performance was analyzed through the dynamic sinusoidal tests under periodic cyclic load. The experimental results exhibited that the S-M-S composite damping structure with SR as reinforcement and MR as matrix has a controllable high stiffness effect, and its damping and vibration reduction characteristics are significantly enhanced, due to the EIS formed between the multi-material interfaces. Moreover, the fatigue characteristics of the laminated structure were investigated, and the dynamic performance stability with the number of periodic load cycles, as the variable was further discussed. The results demonstrated that the S-M-S composite damping structure can effectively reduce the performance instability of traditional MRs and can maintain specific structural characteristics under high-frequency periodic loads.

(2) The effect of parameters related to preparation and working conditions was explored in depth, based on the single-factor control variable method. The results exhibited that the average dynamic stiffness and energy dissipation characteristics of the LCDSs increase gradually with increasing MR matrix density and preload. However, due to the different growth rates, the loss factor increased first and then decreased. In addition, with the increase in the amplitude and frequency, the loss factor of the composite damping structure decreased first and then increased. Therefore, compared to damping structures with single damping components, the stiffness of the multi-material composite damping structure can be controlled by adjusting the above parameters.

## Figures and Tables

**Figure 1 materials-14-00187-f001:**
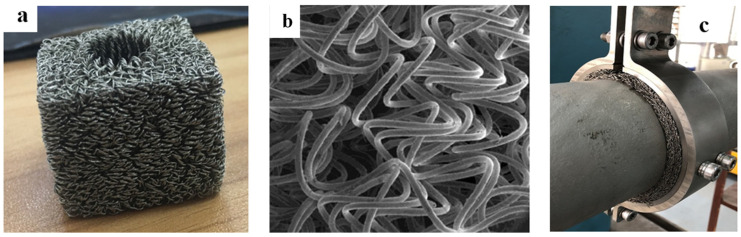
Macro/microstructure of MR and its application on pipeline vibration reduction: (**a**) Macro of MR; (**b**) microstructure of MR; (**c**) Application of MR.

**Figure 2 materials-14-00187-f002:**
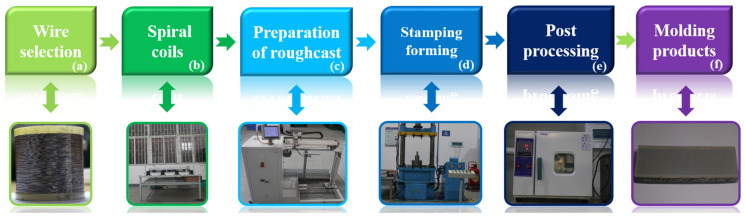
Preparation process of MR.

**Figure 3 materials-14-00187-f003:**
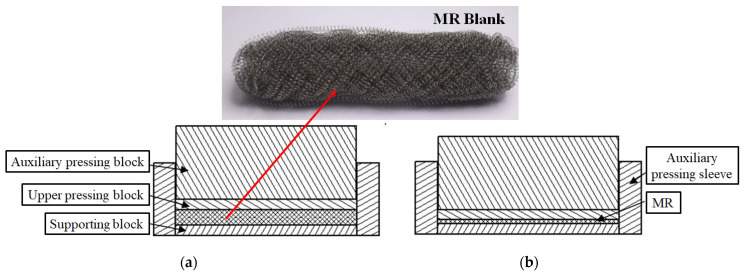
MR stamping forming drawing: (**a**) Initial stamping stage; (**b**) Press forming stage.

**Figure 4 materials-14-00187-f004:**
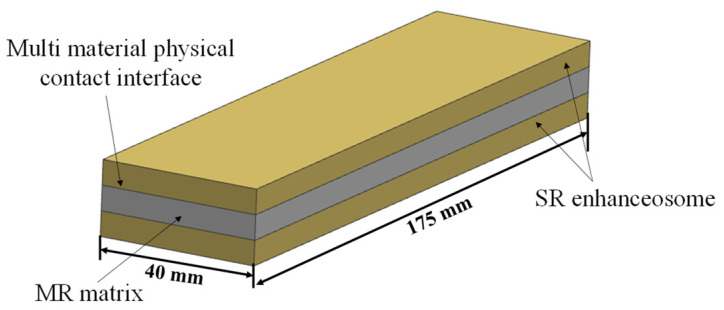
Schematic diagram of the MR- and SR-based LCDS.

**Figure 5 materials-14-00187-f005:**
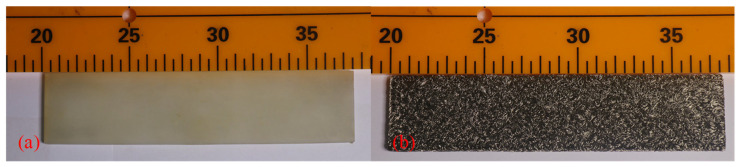
Specimen dimensions: (**a**) specimen of SR; (**b**) specimen of MR.

**Figure 6 materials-14-00187-f006:**
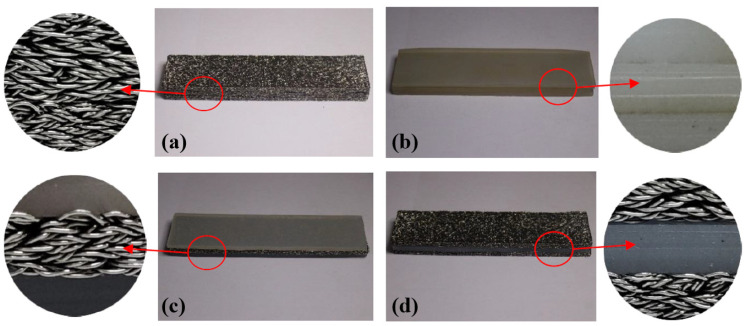
MR/SR laminated structures: (**a**) M-M-M laminated structure; (**b**) S-S-S laminated structure; (**c**) S-M-S laminated structure; (**d**) S-M-S laminated structure.

**Figure 7 materials-14-00187-f007:**
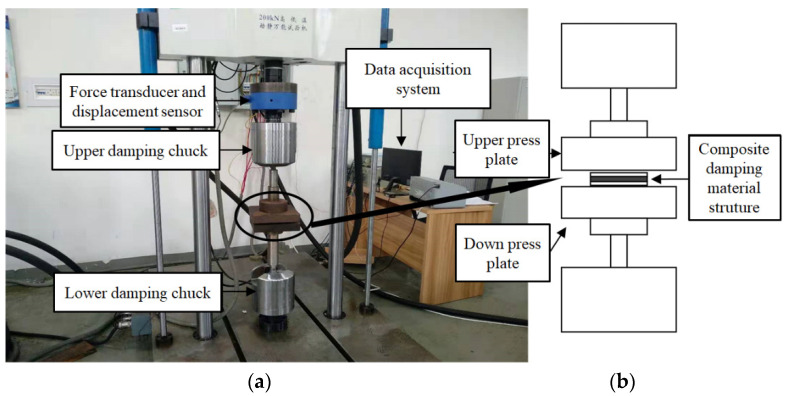
Dynamic testing system: (**a**) Testing system; (**b**) Testing equipment principle.

**Figure 8 materials-14-00187-f008:**
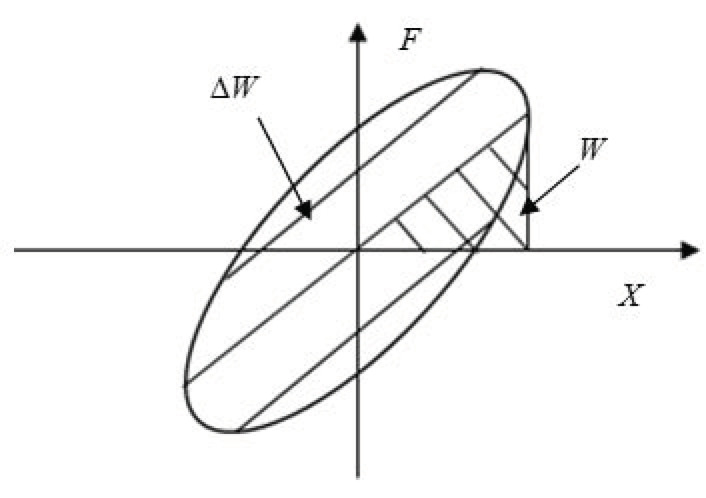
Hysteresis loop.

**Figure 9 materials-14-00187-f009:**
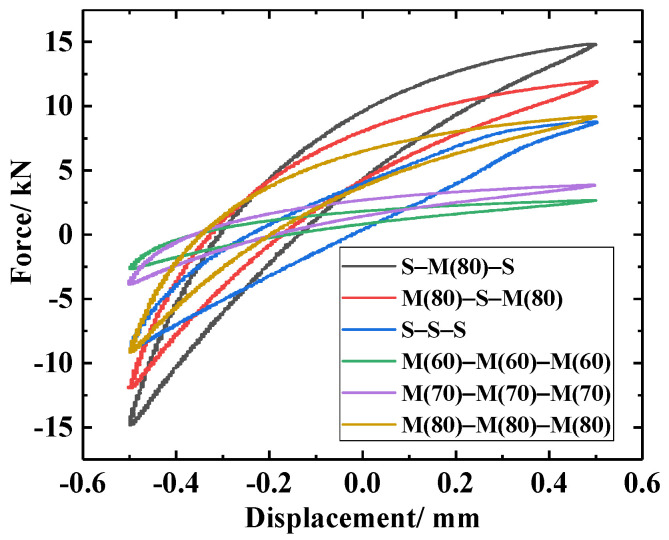
Hysteretic curves of the different laminated structures.

**Figure 10 materials-14-00187-f010:**
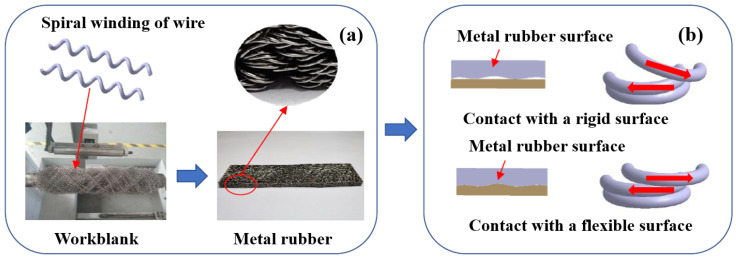
MR forming process and stress state of the metal wires contacting a rigid and a flexible surface, respectively: (**a**) MR forming process; (**b**) Contacting state.

**Figure 11 materials-14-00187-f011:**
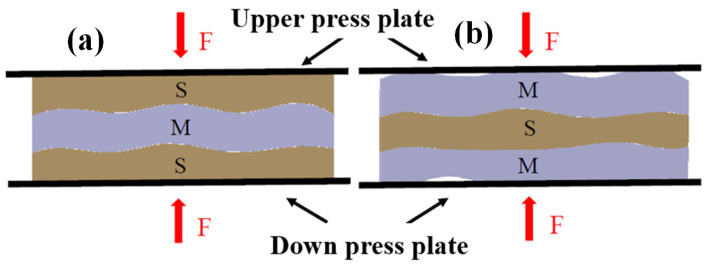
Stress state of the different laminated damping structures: (**a**) S-M-S laminated structure; (**b**) M-S-M laminated structure.

**Figure 12 materials-14-00187-f012:**
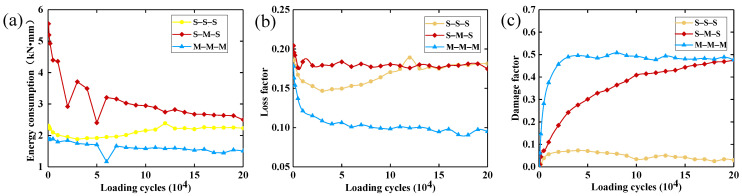
Dynamic performance based on the LCDS: (**a**) Energy consumption; (**b**) Loss factor; (**c**) Damage factor.

**Figure 13 materials-14-00187-f013:**
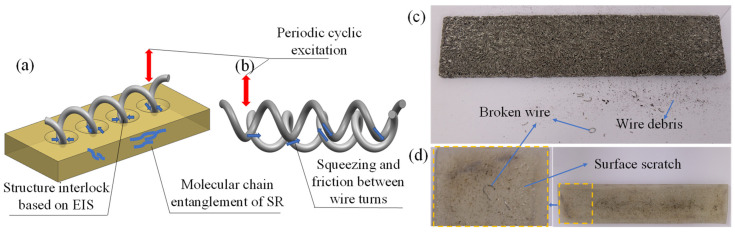
Schematic diagram of the physical contact interface of the different laminated structures and material wear under periodic load: (**a**) Extrusion between MR and SR; (**b**) Extrusion between Wires; (**c**) Wear of MR; (**d**) Wear of SR.

**Figure 14 materials-14-00187-f014:**
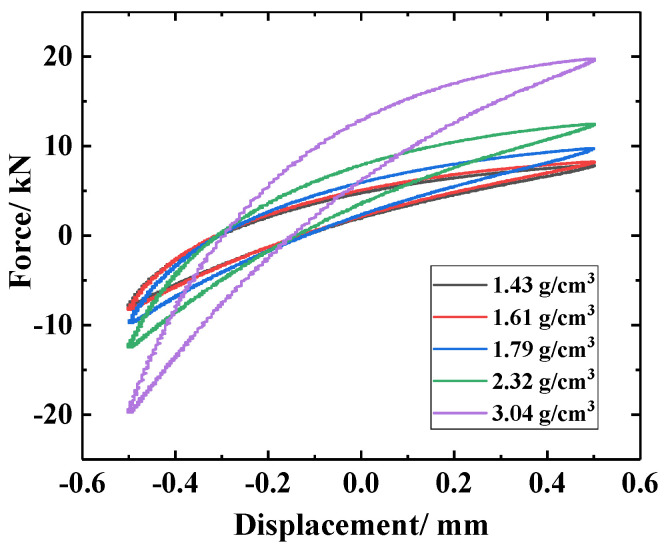
Hysteresis loops with different MR density.

**Figure 15 materials-14-00187-f015:**
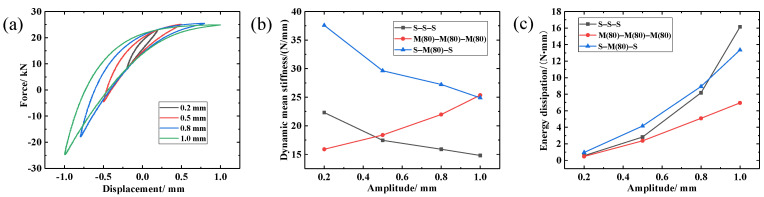
Dynamic performance of LCDS under different amplitudes: (**a**) Hysteresis loops under different amplitudes; (**b**) Variation of dynamic mean stiffness under different amplitudes; (**c**) Variation of energy dissipation under different amplitudes.

**Figure 16 materials-14-00187-f016:**
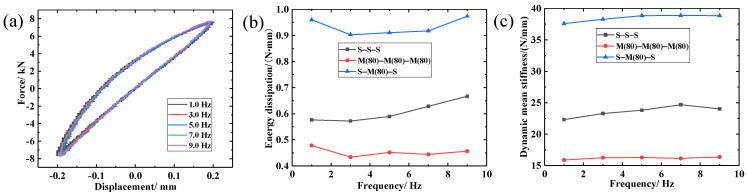
Dynamic performance of LCDS under different frequencies: (**a**) Hysteresis loops under different frequencies; (**b**) Variation of dynamic mean stiffness under different frequencies; (**c**) Variation of energy dissipation under different frequencies.

**Figure 17 materials-14-00187-f017:**
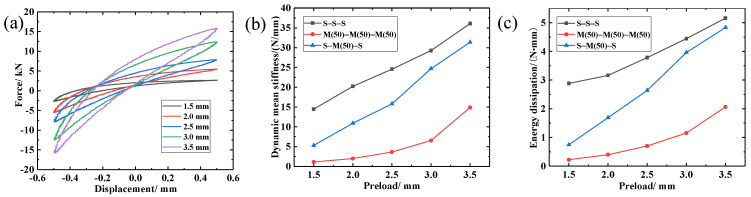
Dynamic performance of LCDS under different preloads: (**a**) Hysteresis loops under different preloads; (**b**) Variation of dynamic mean stiffness under different preloads; (**c**) Variation of energy dissipation under different preloads.

**Table 1 materials-14-00187-t001:** MR specimen parameters.

Element	Density/(g/cm^3^)	Weight/g	Forming Pressure/t
M(40)	1.43	40	13
M(45)	1.61	45	18
M(50)	1.79	50	22
M(60)	2.14	60	40
M(65)	2.32	65	49
M(70)	2.50	70	60

**Table 2 materials-14-00187-t002:** SR and metal wires parameters.

Material	Density (g/cm^3^)	Elastic Modulus (GPa)	Poisson’s Ratio	Applicable Temperature Range (°C)	Height (mm)
Silicon rubber	1.4	1.2	0.48	−30~200	4
304 (06Cr19Ni10)	7.93	193	0.3	−193~800	-

**Table 3 materials-14-00187-t003:** MR preparation parameters.

Wire Diameter (mm)	Coil Diameter (mm)	Helix Pitch (mm)	Winding Angle	Forming Size (mm)
0.2	1.4	1.2	45	175 × 40

**Table 4 materials-14-00187-t004:** Basic parameters of the dynamic testing machine.

Model of Testing Machine	Maximum Load Excitation (kN)	Maximum Loading Displacement (mm)	Loading Frequency Range (Hz)
SDS-200	200	±50	0.01–50

**Table 5 materials-14-00187-t005:** Variation in the damping characteristics of each laminated structure.

Laminated Structure	η	$\bar{K}$	ΔW
S-M(80)-S	0.179	29.640	4.160
M(80)-S-M(80)	0.170	23.686	3.196
S-S-S	0.203	17.465	2.822
M(80)-M(80)-M(80)	0.164	18.393	2.373
M(70)-M(70)-M(70)	0.190	7.751	1.149
M(60)-M(60)-M(60)	0.220	5.340	0.889

**Table 6 materials-14-00187-t006:** Single-factor control test parameters.

Group	Laminated Structure	Density of MR Matrix/(g/cm^3^)	Amplitude /mm	Frequency/Hz	Preload /mm
1	S-M-S	1.43/1.61/1.79/2.32/3.04	0.5	1.0	2.0
2	S-M-S	2.86	0.2/0.5/0.8/1.0	1.0	2.0
3	S-M-S	2.86	0.2	1.0/3.0/5.0/7.0/9.0	2.0
4	S-M-S	1.79	0.5	1.0	1.5/2.0/2.5/3.0/3.5

**Table 7 materials-14-00187-t007:** Dynamic properties based on the density of the MR matrix.

Laminated Structure	Density/(g/cm^3^)	η	$\bar{K}$	ΔW
S-M(40)-S	1.43	0.183	15.614	2.261
S-M(45)-S	1.61	0.184	16.392	2.380
S-M(50)-S	1.79	0.190	19.456	2.901
S-M(65)-S	2.32	0.175	24.837	3.447
S-M(85)-S	3.04	0.164	39.345	5.111

**Table 8 materials-14-00187-t008:** Dynamic parameter variation with amplitude.

Laminated Structure	Amplitude/mm	η	$\bar{K}$	ΔW
S-M(80)-S	0.2	0.199	37.579	0.960
	0.5	0.179	29.640	4.160
	0.8	0.164	27.230	8.935
	1.0	0.171	24.898	13.353

**Table 9 materials-14-00187-t009:** Dynamic parameter variation with frequency.

Laminated Structure	Frequency /Hz	Loss Factor *η*	$\mathrm{Dynamic}\;\mathrm{Average}\;\mathrm{Stiffness}\;\bar{K}$	Energy Consumption Δ*W*
S-M(80)-S	1.0	0.199	37.579	0.960
	3.0	0.196	38.264	0.903
	5.0	0.196	38.844	0.911
	7.0	0.198	38.880	0.918
	9.0	0.204	38.849	0.974

**Table 10 materials-14-00187-t010:** Dynamic parameter variation with preload.

Laminated Structure	Preload/mm	*η*	$\bar{K}$	ΔW
S-M(50)-S	1.5	0.178	5.297	0.745
	2.0	0.196	10.892	1.690
	2.5	0.215	15.785	2.638
	3.0	0.203	24.686	3.962
	3.5	0.195	31.361	4.833

## Data Availability

The data presented in this study are available on request from the corresponding author. The data are not publicly available due to privacy.

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
