# Peer review of "Dynamic Performance of Laminated High-Damping and High-Stiffness Composite Structure Composed of Metal Rubber and Silicone Rubber"

_materials, 2021, doi:10.3390/ma14010187_

Round 1
Reviewer 1 Report
The manuscript discussed the damping performance of the multi-layered metal rubber and silicone rubber composites. Many experiments were carried out and several differences according to changing the parameters were reported. However, there were many unclear points about the experiments. Therefore, major revision before publication would be recommended.
One of the most unclear points was the interface between silicone rubber and metal rubber. Are they bonded together? Or just contacted together?
How was the connection of the specimen to the upper and down press plate? Are they bonded? Or just sandwiches without connection?
These questions come from the force-displacement curves. Because force changed from positive to negative, the specimens were tensiled as well as compressed. I don't know if the positive force means compression or tension, but huge tension was applied to the specimens. Therefore, the authors should clearly explain how the specimens behaved during the tension?
Interface condition also affects fatigue behavior. After the fatigue tests, were the MR and SR still bonded together? Or was the interface fractured and/or wore?
From table 5, S-M(80)-S had Delta_W=4.1595 but S-M(82)-S had 5.111 (table. 7). Just 2% change in density generates 20% increase in Delta_W? Why? Please explain.
In 4.2.2, the amplitude effect was discussed, but there was no information about preload. Why not the authors do not give such important information? Please explain. In addition, if change the preload value, the amplitude effect may changes, which means the preload effect and amplitude effect related together. Why not the authors discuss the preload effect as well in this section?
P13, L389 ==> Table 6?????
Table 9 gives information about S-M(80)-S, but Figure 16 deals with the results of S-M(50)-S, which does not make sense. Please explain.
Author Response
Dear Editors and Reviewers:
On behalf of my co-authors, we thank you very much for giving us an opportunity to revise our manuscript, we appreciate editor and reviewers very much for their positive and constructive comments and suggestions on our manuscript entitled “Dynamic performance of laminated high-damping and high-stiffness composite structure composed of metal rubber and silicone rubber”. We have tried our best to revise our manuscript according to the comments. Attached please find the revised version, which we would like to submit for your kind consideration.
We would like to express our great appreciation to you and reviewers for comments on our paper. Looking forward to hearing from you.
Thank you and best regards.
Yours sincerely,
Zhiying Ren

Reviewer 2 Report
The paper still needs major revision before making a decision:
Revise how to present references according to the journal’s requirement. You cannot mixed both numbers and names.
The written English still needs to be improved in several parts of the document.
I do not think the term “metal rubber” is appropriated here… It is just metal wires entangled. Where is the rubber here ???
There is a problem with Table numbers: they start at 3 on line 141 (text reference) !!! Revise all the document.
Always put a space between values and units. See Figure 4 for example. Revise the whole document.
Line 49: define IPN.
Line 154: the silicon rubber has a modulus of 1.2 GPa !!!! This is very high. Any information on the metal wires properties ?
Table 3: 2.50 on the last line.
Table 3: foaming pressure ??? How the material is foamed ? I do not see any porosity analysis. This is not clear…
Table 4 : kN
There is not enough information on how the fatigue tests were performed: conditions, repetition, type of tests, sample dimensions, etc… The part in Section 4.1 is misplaced and incomplete !
Table 5: the samples reported here do not correspond with the ones presented before (codes)…
Table 5: too many digits reported. Three is enough (any error estimation ?).
Figure 12 is misleading. What do you mean by “loading times” ? Usually the number of cycle is the independent parameter. This does not fit with the description of lines 312-313 !
Figure 12: why the damage factor does not go up to 1.0 : failure to report the fatigue life time Nf ? Failure analysis should be also reported for fatigue analysis.
Table 6: two decimals for density everywhere !
Table 7: too many digits reported. Three is enough (any error estimation ?).
Pages 12-13: is S-M(80)-S the same as S-M(82)-S ?
Table 8: too many digits reported. Three is enough (any error estimation ?).
Table 9: too many digits reported. Three is enough (any error estimation ?).
Table 10: too many digits reported. Three is enough (any error estimation ?). Why use the same sample (S-M(80)-S) for all the comparison ?
Author Response
Dear Editors and Reviewers:
On behalf of my co-authors, we thank you very much for giving us an opportunity to revise our manuscript, we appreciate editor and reviewers very much for their positive and constructive comments and suggestions on our manuscript entitled “Dynamic performance of laminated high-damping and high-stiffness composite structure composed of metal rubber and silicone rubber”. We have tried our best to revise our manuscript according to the comments and reached please find the revised version, which we would like to submit for your kind consideration.
We would like to express our great appreciation to you and reviewers for comments on our paper. Looking forward to hearing from you.
Thank you and best regards.
Yours sincerely,
Corresponding author:
Name: Zhiying Ren

Round 2
Reviewer 1 Report
First of all, why were there so many mistakes about experimental data in the submitted manuscript? Is it a level just to conclude with a single word of negligence?
Are you really sure that the data are 100% correct? I cannot believe the data in the manuscript are correct after seeing hundreds of mistakes in the 1st submitted ver.
Response 1,2 and 3 were not reflected in the manuscript, which means I have to ask the same question because "responds to the reviewers" is not open to the public.
Especially, the definition of the force (what the zero point means, negative means more compressed or less, etc) should be clearly identified in the manuscript. Otherwise, everybody thinks zero force means zero pressure (no compression nor tension), and nobody knows what the positive force means.
In addition, the force was not zero at zero displacements in many cases. Please explain in the manuscript what this means.
About Response 4 "the wear is not serious"
How can the reader judge about it? Please show the evidence even if it was not serious.
About Response 6 "preload and amplitude are two unrelated variables"
Comparing a highly preloaded sample with a small amplitude (which means average preload is extremely high) and an almost zero-preloaded sample with a small amplitude (which means almost zero average preload), I think the movability of the wire of MR is totally different. Therefore, even with the same amplitude, there is a possibility that the movement of the wire changes. Also, the friction force that arose between the wires could be different.
If there is any evidence, please show it in the manuscript. If not, please clearly explain from the mechanical point of view why the authors think "preload" and "amplitude" are independent together.
Author Response
Dear Editors and Reviewers:
On behalf of my co-authors, we thank you very much for giving us an opportunity to revise our manuscript, we appreciate editor and reviewers very much for their positive and constructive comments and suggestions on our manuscript entitled “Dynamic performance of laminated high-damping and high-stiffness composite structure composed of metal rubber and silicone rubber”. We have tried our best to revise our manuscript according to the comments. Attached please find the revised version, which we would like to submit for your kind consideration.
We would like to express our great appreciation to you and reviewers for comments on our paper. Looking forward to hearing from you.
Thank you and best regards.
Yours sincerely,
Zhiying Ren
List of Responses
Dear Editors and Reviewers:
Thank you for your letter and for the reviewers’ comments concerning our manuscript entitled “Dynamic performance of laminated high-damping and high-stiffness composite structure composed of metal rubber and silicone rubber”. Those comments are all valuable and very helpful for revising and improving our paper, as well as the important guiding significance to our researches. We have studied comments carefully and have made correction, which we hope meet with approval.
Responds to the Reviewer 1:
Comment 1: First of all, why were there so many mistakes about experimental data in the submitted manuscript? Is it a level just to conclude with a single word of negligence?
Are you really sure that the data are 100% correct? I cannot believe the data in the manuscript are correct after seeing hundreds of mistakes in the 1st submitted ver.
Response 1: The data in the manuscript is wrong due to our negligence,which is indeed our responsibility, and we are deeply sorry about it. Our date changes appear in line 344(table 6)、line 352(table 7)、line 400 S-M(80)-S (Modify here to see Table 9).
Secondly, we are sorry for our improper marking, which may cause your misunderstanding. The other data changes in the table are due to the second reviewer's request to unify the number of decimal points.
Comment 2: Response 1, 2 and 3 were not reflected in the manuscript, which means I have to ask the same question because "responds to the reviewers" is not open to the public.
Comment a: One of the most unclear points was the interface between silicone rubber and metal rubber. Are they bonded together? Or just contacted together?
Response a: Thanks for the referee’s kind suggestion. Corrections have been done in line 106 in yellow mark.
“This kind of structure just stacks different materials together, instead of bonding different materials by chemical or physical methods
Comment b: How was the connection of the specimen to the upper and down press plate? Are they bonded? Or just sandwiches without connection?
Response b: Thanks for the referee’s kind suggestion. Corrections have been done in line 188-190 in yellow mark.
Comment 3: Especially, the definition of the force (what the zero point means, negative means more compressed or less, etc) should be clearly identified in the manuscript. Otherwise, everybody thinks zero force means zero pressure (no compression nor tension), and nobody knows what the positive force means.
Response 3: Thanks for the referee’s kind suggestion. Corrections have been done in line 190-193 in yellow mark.
“During the experiment, there is a certain displacement to preload to the sample. The position at this time is the zero point. When moving downward from this zero point, the pressure is negative, and when moving upward from the equilibrium point, the pressure is positive. The sample only compressed during the whole experiment.”
Comment 4: In addition, the force was not zero at zero displacements in many cases. Please explain in the manuscript what this means.
Response 4: In the experiment, the instrument is controlled by controlling the displacement of the upper beam, so the displacement is symmetrical. If the force is asymmetric, it will be difficult to observe when comparing in the figure. Therefore, in order to make a better comparison, we also symmetrical the force. This method no effect on the experimental results, just for better comparison. Therefore, it happened that the force was not zero at zero displacements in many cases. This method can also be seen in other literatures, such as Fig.13、Fig.16、 Fig 18 in the literature[1]. Fig.2、Fig.8、Fig.9 in the literature[2].
[1]Xue. X. et al. Manufacture Technology and Anisotropic Behaviour of Elastic-Porous Metal Rubber [J]. International Journal of Lightweight Materials and Manufacture, 2019, 3(2).
(https://doi.org/10.1016/j.ijlmm.2019.08.005).
[2]Ren. Zhi. Ying. et al. Study on Damping Energy Dissipation Characteristics of Cylindrical Metal Rubber in Nonforming Direction[J]. Advances in Materials Science and Engineering, 2018, 2018(11):1-10. (https://doi.org/10.1155/2018/5014789)
Comment 5: About Response 4 "the wear is not serious"
How can the reader judge about it? Please show the evidence even if it was not serious.
Response 5: Thanks for the referee’s kind suggestion. Corrections have been done in line 314-317 in yellow mark.
“As shown in Fig. 13(c), a certain amount of debris and metal broken wire are produced under periodic load. Due to the friction of metal wire, the surface wear is formed on the surface of SR as showed in Fig. 13(d)). At the same time, it can be seen that the depth of wear is not very deep.”
Comment 6: About Response 6 "preload and amplitude are two unrelated variables"
Comparing a highly preloaded sample with a small amplitude (which means average preload is extremely high) and an almost zero-preloaded sample with a small amplitude (which means almost zero average preload), I think the movability of the wire of MR is totally different. Therefore, even with the same amplitude, there is a possibility that the movement of the wire changes. Also, the friction force that arose between the wires could be different. If there is any evidence, please show it in the manuscript. If not, please clearly explain from the mechanical point of view why the authors think "preload" and "amplitude" are independent together.
Response 6: In the actual application process, it is often given a certain preload to ensure the stability of the pipeline system (Fig. 1. C), and then consider the actual applied amplitude. So we discuss preload and amplitude separately. The same comparison can be found in some literature. For example, Wang et al[1] discussed the change of the tangent module and the loss factor of MR-CHC sandwich structure with applied amplitude for given a preload and excitation frequency in part 4.3.4. (4.3.4. Effect of the applied amplitude). And discussed the effect of preload level in part 4.3.5(4.3.5. Effect of the pre-compression level). Wang hong et al[2] conducted the effects of amplitude (4.1 Effect of Dynamic Strain Amplitude) and pre-compression (4.2 Effect of Pre-Compression) of MR damper on the vibration reduction performance respectively. Zou et al[3] also conducted the influence of preloading and vibration level.
[1]Yong Jing Wang, et al. Experimental investigation on enhanced mechanical and damping performance of corrugated structure with metal rubber. Thin-Walled Structures 2020. 154. 106816 (https://doi.org/10.1016/j.tws.2020.106816)
[2]WANG Hong, et al. Nonlinear static and dynamic properties of metal rubber dampers [R]. Leuven, Belgium: International Conference on Noise and Vibration Engineering (ISMA), 2010
[3] Zou. Guang. Ping, et al. Effects of preloading and vibration level on the vibration characteristics of metal rubber damper [J]. Journal of Vibration and Shock. 2015. 34(22):173-177+191. (10.13465/j.cnki.jvs.2015.22.030).

Reviewer 2 Report
ok
Author Response

(The authors gave the same response as above.)

Round 3
Reviewer 1 Report
Just a short comment about response 6:
If there was an objective for a certain discussion, indicating the reasons, such as "it is important when considering the application to pipelines", would be helpful for the readers to easily understand the importance. Because the phenomena itself is very complicated and there exist many indistinguishable effects, many questions arise if there are no indications about the objectives of the discussion.
Please add the information written in response 6 to the manuscript.
Author Response
List of Responses
Dear Editors and Reviewers:
Thank you for your letter and for the reviewers’ comments concerning our manuscript entitled “Dynamic performance of laminated high-damping and high-stiffness composite structure composed of metal rubber and silicone rubber”. Those comments are all valuable and very helpful for revising and improving our paper, as well as the important guiding significance to our researches. We have studied comments carefully and have made correction, which we hope meet with approval.
Responds to the Reviewer 1:
Comment 1: If there was an objective for a certain discussion, indicating the reasons, such as "it is important when considering the application to pipelines", would be helpful for the readers to easily understand the importance. Because the phenomena itself is very complicated and there exist many indistinguishable effects, many questions arise if there are no indications about the objectives of the discussion.
Please add the information written in response 6 to the manuscript.
Response 1: Thanks for the referee’s kind suggestion. Your suggestion has provided greatful help to the improvement of our article. Thank you again for your help. Corrections have been done in line 426-429.
“Amplitude and preload are the main factors affecting porous materials, and there is a certain correlation. However, in the actual application process, it is often given a certain preload to ensure the stability of system, and then consider the actual applied amplitude[25]. Therefore, we discuss the influence of amplitude and preload on the LCDS separately.”

This manuscript is a resubmission of an earlier submission. The following is a list of the peer review reports and author responses from that submission.
Round 1
Reviewer 1 Report
This paper reports an experimental study on the damping properties of three-layer materials made from a rubber (silicone) and metal (stainless steel). The information could be of interest, but there are several points limiting the understanding of the work done and the significance of the results reported. Here is a list of comments based on the paper analysis:
The paper should state that only symmetric structures were studied (see Figure 3 but the images are too small, especially the ones in the middle). But asymmetric structures would also be of interest: s-s-m vs. m-m-s.
The properties of a single layer must be reported before understanding the properties of the multi-layer ones !
Figure 1 is missing !
Very limited information given on the raw materials used, and even less information on the samples preparation method… So it is difficult to understand the work done. I still do not understand the definition of a “metal rubber” !!!
The experiments are not well described: What kind of mechanical test was performed (compression ?) and what was the conditions (small amplitude ?). Presentation order must be improved.
Table 2: too many digits reported (3 is enough). But based on Table 1, several more possibilities can be investigated. Why not done here ?
Figure 4: again very small, especially the font size (difficult to read).
The paragraph above Table 4 is impossible to understand. It must be completely re-written.
Tables 4 and 5 and 6 and 7: too many digits reported (they must be uniform too). The values are “changes/variation” compare to what ?
The same problem occur is the figures with captions involving “Variation”. This is not appropriate… Please rephrase the captions.
Figures 5 and 7: any error bars to report (how many repetitions for each test ?) ? Within experimental uncertainty, I do not believe that significant effects are obtained, especially in Figure 7…
Figure 6 is missing !
Line 279: please make a list of all the mechanisms involved.
Line 310: there is no Figure 18 ! Numbers must be re-checked for figures.
Lines 331-333: I do not understand this part.
Table 9 is not clear: is this a statistical analysis of the results ? More details is requested.
What is a “blank” (everywhere) ?
What is “pre-tightness” (line 192 for example) ?
Line 195 : “1.5 mm” is not a load or preload (it is a deformation) !!!
The presentation must be improved (see Table 1 and line 114 for example).
Revise the use of “et al.” and how to present references in the manuscript.
Revise for spaces, superscripts and capital letters throughout the manuscript.
The written English can be polished at some places.
Reviewer 2 Report
The manuscript discusses the damping performance of the metal rubber/silicone rubber composites. Interesting laminated structures are introduced, but some corrections should be required before publication.
1. Figure 1 is missing.
2. Although it is said that the metal rubber has a narrow vibration reduction frequency band, the authors should show frequency values in which frequency range vibration reduction is good and/or weak for the metal rubber. Does the experimental condition well fit the weak frequency range?
3. Do the introduced composites have a good vibration reduction at the high-frequency range? The authors should show that the good point of the metal rubber is not lost with the composites.
4. Although there is an explanation about the metal rubber and the silicone rubber, no explanation was found about how to create a laminated structure. Was the silicone rubber glued to the metal rubber? Or just was put onto it? The contact condition between the silicone rubber and the metal rubber is very important for the vibration results. Therefore, a detailed explanation about the effect of the contact condition on the experimental results should be discussed.
5. Additionally, there is uncertainness about the durability of the laminated composites. Because the hardness of the metal rubber and the silicone rubber is different and failure, abrasion, friction and so on at the interface between them should occur, the authors should show performance change for long time vibration.
6. Significant digits are not unified at Tables 1, 3, 5, 6, 7, 8, and 9.